# Structural and functional insights into a novel two-component endolysin encoded by a single gene in *Enterococcus faecalis* phage

Biao Zhou[1�], Xiangkai Zhen[1,2�], Huan Zhou[3�], Feiyang Zhao[4�], Chenpeng Fan[5], Vanja Perčulija (ID)[1], Yigang Tong[4], Zhiqiang Mi[6]*, Songying Ouyang (ID)[1,2]*

1 The Key Laboratory of Innate Immune Biology of Fujian Province, Provincial University Key Laboratory of Cellular Stress Response and Metabolic Regulation, Biomedical Research Center of South China, Key Laboratory of OptoElectronic Science and Technology for Medicine of the Ministry of Education, College of Life Sciences, Fujian Normal University, Fuzhou, China, 2 Laboratory for Marine Biology and Biotechnology, Pilot National Laboratory for Marine Science and Technology (Qingdao), Qingdao, China, 3 Shanghai Institute of Applied Physics, Chinese Academy of Sciences, Shanghai, China, 4 Beijing Advanced Innovation Center for Soft Matter Science and Engineering (BAIC-SM), College of Life Science and Technology, Beijing University of Chemical Technology, Beijing, China, 5 Department of Biochemistry and Molecular Biology, School of Basic Medical Sciences, Wuhan University, Wuhan, China, 6 State Key Laboratory of Pathogen and Biosecurity, Beijing Institute of Microbiology and Epidemiology, Beijing, China

y These authors contributed equally to this work.
* zhiqiangmi_ime@163.com (ZM); ouyangsy@fjnu.edu.cn (SO)

**Data Availability Statement:** All data needed to evaluate the conclusions in the paper are present in the paper and/or the Supporting Information. Coordinates and structure factors are deposited in

## Abstract

Using bacteriophage-derived endolysins as an alternative strategy for fighting drug-resistant bacteria has recently been garnering renewed interest. However, their application is still hindered by their narrow spectra of activity. In our previous work, we demonstrated that the endolysin LysIME-EF1 possesses efficient bactericidal activity against multiple strains of *Enterococcus faecalis* (*E. faecalis*). Herein, we observed an 8 kDa fragment and hypothesized that it contributes to LysIME-EF1 lytic activity. To examine our hypothesis, we determined the structure of LysIME-EF1 at 1.75 Å resolution. LysIME-EF1 exhibits a unique architecture in which one full-length LysIME-EF1 forms a tetramer with three additional C-terminal cell-wall binding domains (CBDs) that correspond to the abovementioned 8 kDa fragment. Furthermore, we identified an internal ribosomal binding site (RBS) and alternative start codon within *LysIME-EF1* gene, which are demonstrated to be responsible for the translation of the truncated CBD. To elucidate the molecular mechanism for the lytic activity of LysIME-EF1, we combined mutagenesis, lytic activity assays and *in vivo* animal infection experiments. The results confirmed that the additional LysIME-EF1 CBDs are important for LysIME-EF1 architecture and its lytic activity. To our knowledge, this is the first determined structure of multimeric endolysin encoded by a single gene in *E. faecalis* phages. As such, it may provide valuable insights into designing potent endolysins against the opportunistic pathogen *E. faecalis*.

the Protein Data Bank with the PDB entries: 6IST (WT LysIME-EF1) and 6L00 (LysIME-EF1 CBD).

**Funding:** This work was partially supported by the National Natural Science Foundation of China grants (31770948, and 31570875 to S. O.; 31800159 to X. Z., 81572045 to Z. M.), The high-level personnel introduction grant of Fujian Normal University (Z0210509) to S. O., Natural Science Foundation of Fujian Province (2019J05065) to X. Z.. We thank the support of the scientific research innovation program "Xiyuanjiang River Scholarship" of College of Life Sciences, Fujian Normal University to X. Z.. The funders had no role in study design, data collection and analysis, decision to publish, or preparation of the manuscript.

**Competing interests:** The authors have declared that no competing interests exist.

## Author summary

LysIME-EF1, an endolysin that lyses *E. faecalis*, displays the prospect of treating *E. faecalis* infection. We find that the C-terminal cell-wall binding domain (CBD) is important for the lytic activity of LysIME-EF1. By determining the crystal structures of wild type (WT) LysIME-EF1 and its C-terminal CBD, this study reveals how the holoenzyme is organized to carry out its highly efficient lytic activity. Our finding provides structural and functional evidence that LysIME-EF1 belongs to a unique two-component multimeric endolysin encoded by a single gene.

## Introduction

*Enterococcus faecalis* (*E. faecalis*) is a Gram-positive bacterium that commonly inhabits oral cavity, lower intestinal tract, and vaginal tract of healthy humans or other mammals [1]. However, *E. faecalis* is also a major opportunistic pathogen that causes a large number of community-acquired and hospital infections, where high levels of antibiotic resistance in *E. faecalis* increase its pathogenicity [2]. *E. faecalis* is particularly life-threatening for immunocompromised patients, and is known to cause endocarditis, bacteremia, urinary tract infections, meningitis etc. In addition, *Enterococcus* has been found related to root canal-treated teeth and diabetic foot infections [3–6].

Endolysins are hydrolytic enzymes that specifically recognize the host cell-wall and cleave the peptidoglycan to digest the bacterial cell-wall for release of phage progeny during the lytic cycle [7, 8], which is why they hold promise for the treatment of multidrug-resistant bacterial infections as an alternative to antibiotics [9–11]. Generally, most endolysins are composed of a single polypeptide chain divided into an N-terminal catalytic domain (CD) and a C-terminal cell-wall binding domain (CBD), which are interconnected by a short flexible linker [12–14]. Some endolysins contain single CD and some are composed of multiple CDs linked to CBD [15–17]. Catalytic domains are responsible for cleavage of peptidoglycan bonds within the cell-wall and can be classified into four groups: N-acetylglucosaminidases, N-acetylmuramoyl-L-alanine amidases, N-acetylmuramidases (lysozymes), and endopeptidases [18–20]. In contrast, CBDs display much greater variety and distinguish discrete epitopes present within the host cell-wall, typically carbohydrates or teichoic acids, thus giving rise to species-specific or strain-specific activity of a particular endolysin [21, 22]. The modular structure of lysins makes it possible to design bioengineered endolysins that have desired properties, such as higher activity, or broader killing spectrum. Chimeric endolysins such as ClyH and ClyJ are engineered to fight against methicillin-resistant *Staphylococcus aureus* (MRSA) and multidrug-resistant *Streptococcus pneumoniae* (MRSP) [23, 24].

Although most known endolysins are composed of a single polypeptide chain containing one CD and one CBD [25–27], some exceptions are known. For example, in some cases two or three CDs are linked to a single CBD made of two separated functional regions [15, 28, 29]. Moreover, PlyC, a unique multimeric endolysin from streptococcal C1 phage, consists of two components encoded by two separated genes located in one single operon [30]. The structure of PlyC comprises an enzymatic component (PlyCA) associated with eight copies of the cell-wall binding component (PlyCB) [31]. Interestingly, recent studies reported the existence of multimeric endolysins encoded by one single gene, such as Lys170 from enterococcal phage F170/08 [32], CTP1L that targets *Clostridium tyrobutyricum*, and CD27L that targets *Clostridium difficile* [33, 34].

LysIME-EF1, an endolysin isolated from *E. faecalis* phage possesses a highly efficient lytic activity to multidrug resistance *E. faecalis* [35]. In this study, we found that LysIME-EF1 possesses efficient bactericidal activity and can lyse 29 *E. faecalis* clinical strains, including the lethal vancomycin-resistant *Enterococcus faecalis* 002 (VREF). Furthermore, the discovery that a truncated CBD fragment is co-purified with the full-length LysIME-EF1 prompted us to solve the crystal structures of LysIME-EF1 holoenzyme and the CBD, which were determined both at 1.75 Å resolution. These structures and the accompanying biochemical experiments and *in vivo* animal infection experiments provided insights into the molecular mechanism of LysIME-EF1 lytic activity.

## Results

### Expression of the *LysIME-EF1* gene appears to result in two stable polypeptides

In our previous work, we isolated an endolysin derived from *E. faecalis* bacteriophage [35]. PSI-BLAST search predicted that LysIME-EF1 consists of an N-terminal catalytic CHAP (cysteine, histidine-dependent amidohydrolases/peptidases) domain and a C-terminal CBD (cell-wall binding domain) (**Fig 1A**). Remarkably, LysIME-EF1 displays efficient bactericidal activity against 29 clinical strains of *E. faecalis* (**S1A Fig**). Subsequently, we found that expression of full-length N-terminally linked to a 6xHis-tag LysIME-EF1 in *E. coli* resulted in a strongly co-purified polypeptide with the approximate molecular mass of 8 kDa (**Fig 1B**). The 8 kDa band was overlooked in our previous works because of relatively longer running time for SDS-PAGE [35]. To identify this protein, we conducted N-terminal sequencing and found that the first four amino acid residues coincided with the sequence of $M_{168}F_{169}I_{170}Y_{171}$, which corresponds to the middle of LysIME-EF1. These results suggest that the 8 kDa fragment is the CBD of LysIME-EF1 rather than a product of protein degradation.

To further examine the hypothesis that the extra CBD fragment is incorporated into protein complex, size-exclusion chromatography was performed to analyze the solution state of LysIME-EF1. LysIME-EF1 CBD (8 kDa) and full-length LysIME-EF1 (30 kDa) were co-eluted at 15.5 ml, with the estimated molecular mass of approximately 60 kDa (**S2A Fig**). To obtain the exact molecular mass, analytical ultracentrifugation (AUC) was performed, indicating that LysIME-EF1 exhibited a molecular mass of 55.6 kDa, higher than the theoretical molecular mass of LysIME-EF1 alone (**S2B Fig**). Taken together, these results suggested that the truncated CBD fragments may form a complex with full-length LysIME-EF1.

To assess the effect of additional CBDs on LysIME-EF1 lytic activity, we tested the bactericidal activity of the wild type (WT) LysIME-EF1, LysIME-EF1 CHAP, LysIME-EF1 CBD and mixed WT LysIME-EF1 and LysIME-EF1 CBD at different molar ratios. All the constructs were expressed in *E. coli* and purified as soluble recombinant proteins (**S3A and S3B Fig**). The results revealed that WT LysIME-EF1 has efficient bactericidal activity against *E. faecalis*. In contrast, CHAP alone, CBD alone or a mixture of these two polypeptides exhibited very low bactericidal activity, even at high concentrations (10 μM) (**Fig 1C**). Interestingly, addition of the CBD mixed with full-length LysIME-EF1 at different molar ratios showed that lytic efficiency is inversely proportional to the molar ratio of CBD fragment to full-length LysIME-EF1 (**Fig 1D**).

### Overall structure of WT LysIME-EF1 and the CBD fragment

To better understand the role of the CBD in WT LysIME-EF1 lytic activity, we aimed to determine the structure of LysIME-EF1. Thus, full-length LysIME-EF1 was cloned into the

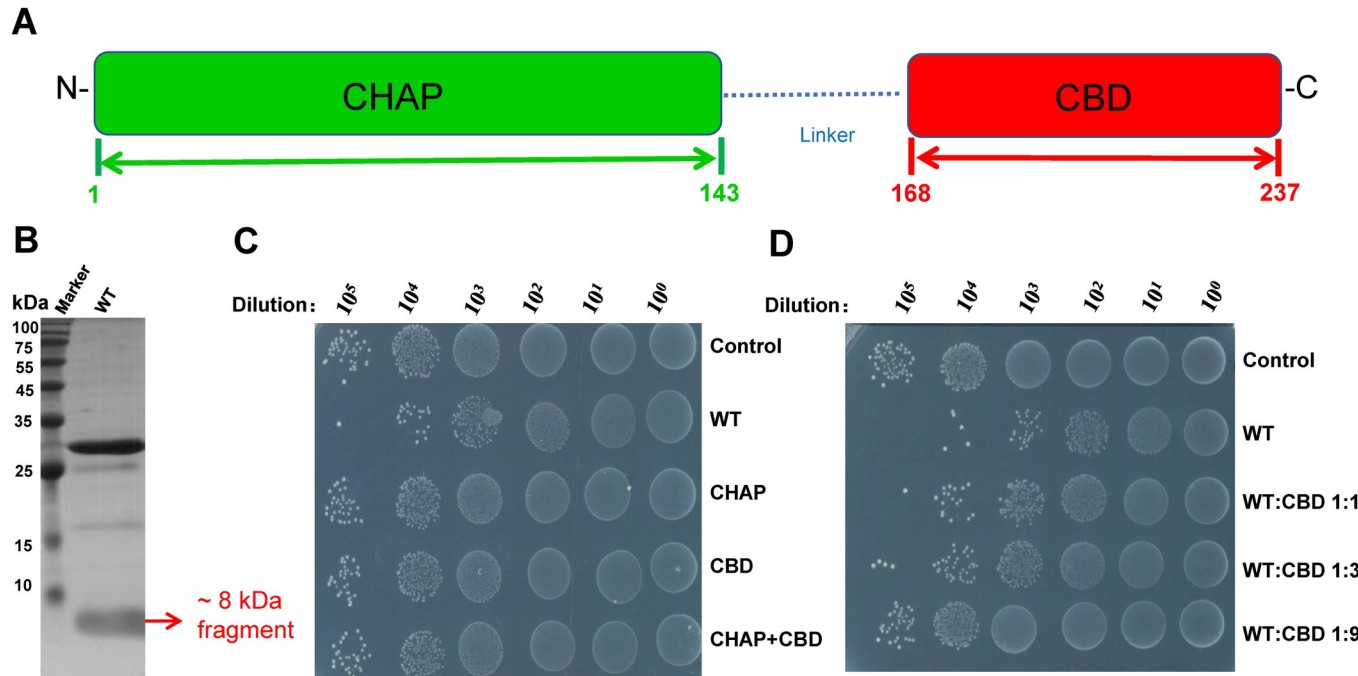

**Fig 1. Expression of *LysIME-EF1* gene appears to results in production of two stable polypeptides. (A)** Domain organization of LysIME-EF1. The full-length LysIME-EF1 contains two domains, i.e. the N-terminal CHAP domain (residues 1–143, colored green) and the C-terminal CBD (residues 168–237, colored red). The linker (residues 144–167) interconnecting the two domains is shown as a dot-dashed blue line. **(B)** The SDS-PAGE analysis of the expression of wild type (WT) LysIME-EF1. **(C)** Survival test of *E. faecalis* 002 on LB agar dishes after the cells were lysed by WT LysIME-EF1 (0.1 μM) as well as CHAP domain protein alone (0.1 μM) or CBD alone (0.1 μM) or mixture of CHAP domain protein plus CBD (10 μM) at 37°C for 2 hours. **(D)** Survival test of *E. faecalis* 002 on LB agar dishes after the cells were lysed by WT LysIME-EF1 (0.1 μM) that mixed with CBD at different molar ratio at 37°C for 2 hours.

expression vector pCold I, expressed in *E. coli* and subsequently purified using Ni affinity chromatography and size-exclusion chromatography. Following extensive crystallization trials, we obtained crystals that diffracted well in the space group *I*422, with cell parameters a = b = 59.88 Å, c = 87.01 Å, α = β = γ = 90.00° (**Table 1**). However, we failed to solve the structure by molecular replacement using the CHAP domain of LysGH15 (PDB entry: 4OLK) as search model. Fortunately, we obtained diffractive crystals of selenium-labeled protein in different crystallization conditions, which were then optimized and diffracted well in the space group *P*12₁1 with cell parameters a = 47.03 Å, b = 56.92 Å, c = 91.52 Å, α = γ = 90.00°, β = 96.29° (**Table 1**). Finally, the structure was solved successfully using the data sets belonging to the space group *P*12₁1 by selenium single-wavelength anomalous dispersion (Se-SAD) method. Interestingly, the structure of the abovementioned datasets in the space group *I*422 was solved later by molecular replacement and determined as LysIME-EF1 CBD homotetramer (**Table 1**).

The crystal structure of WT LysIME-EF1 refined to 1.75 Å resolution revealed that WT LysIME-EF1 possesses a CHAP domain (residues 1–143) and a CBD (residues 168–237), which are connected by a linker (residues 144–167) (**Fig 2A**). The electron densities for the CHAP and the CBD domains are unambiguously defined, whereas the electron density for the linker is missing due to its high flexibility. In addition to the full-length LysIME-EF1, three additional CBD fragments were successfully built into the final model (**Fig 2A**). The four CBDs together form a tetrameric ring 45 Å in length and 45 Å in width. One side of LysIME-EF1 tetrameric CBDs, which we speculate to be responsible for recognition of

**Table 1. X-ray data collection and refinement statistics.**

| Dataset | Se-Met WT LysIME-EF1 | LysIME-EF1 CBD |
|---|---|---|
| **Data collection** | | |
| Wavelength (Å) | 0.9792 | 0.9792 |
| Space group | $P12_11$ | $I422$ |
| Cell dimensions<br>a, b, c (Å)<br>α, β, γ (˚) | 47.03, 56.92, 91.52<br>90.00, 96.29, 90.00 | 59.88, 59.88, 87.01<br>90.00, 90.00, 90.00 |
| Resolution range(Å) | 46.75–1.75(1.813–1.75) | 42.34–1.75(1.813–1.75) |
| unique reflections | 47621(4713) | 8236(552) |
| Rmerge | 0.103 | 0.089 |
| CC1/2 | 0.995 | 0.985 |
| $I/\sigma(I)$ | 16.5(2.1) | 24.3(2.3) |
| Completeness (%) | 97.83(97.42) | 98.81 (92.48) |
| **Refinement** | | |
| Resolution (Å) | 1.75 | 1.75 |
| Completeness (%) | 97.83(97.42) | 99.0(92.8) |
| Rwork | 0.1793(0.3173) | 0.2082 (0.2986) |
| Rfree | 0.2163(0.3674) | 0.2334 (0.3582) |
| No. atoms<br>ion | 3397<br>1 | 582<br>0 |
| Ramachandran plot (%)<br>Favored region<br>Allowed region<br>Outliers region | 99.52<br>0.48<br>0.00 | 100.00<br>0.00<br>0.00 |

Values in parentheses are for highest-resolution shell.

peptidoglycan, is arranged in a planar surface, while the other side is more convex in shape (right panel in **Fig 2B**).

The N-terminal CHAP domain of LysIME-EF1 folds into a typical globular domain containing two α-helices, two $3_{10}$-helices and six β-strands (**Fig 3A**). Dali search [36] against the PDB database indicates that the most closely related structures are those of the CHAP domains of staphylococcal phage-derived endolysins LysGH15 (PDB: 4OLK) and LysK (PDB: 4CSH) (**S4C Fig**) [15, 29]. Structural alignment of the CHAP domains of LysIME-EF1 and LysGH15 shows that the two α-helices and the six β-strands of both proteins are aligned well with each other with an average root mean square deviation (RMSD) of 0.717 Å and 0.742 Å, respectively (**S4A and S4B Fig**). Analogously to LysGH15, LysIME-EF1 CHAP domain also contains a calcium-binding site and a classical amidohydrolase catalytic triad (C29, H90 and N110) (**Fig 3B**). Electron density of the calcium ion is clearly identified in the crystal structure, and the calcium ion is coordinated by the conserved residues D20, D22, W24, G26, D31 and one water molecule (**Fig 3B and 3C**). EDTA completely inactivates the lytic activity of LysIME-EF1, which can only be restored by addition of calcium ions (**Fig 3D**). Similar to LysGH15 [15], mutation of the catalytic residues to alanine reduced the bactericidal activity of LysIME-EF1 (**Fig 3E**).

Ultimately, the dataset diffracted in the space group $I422$ was proved to be LysIME-EF1 CBD and the structure was solved by molecular replacement method at 1.75 Å resolution using the CBD from WT LysIME-EF1 as a search model. There was one molecule in an asymmetric unit. The CBD comprises a four-stranded β-sheet capped on each side by one α-helix (**S5A and S5B Fig**). Crystallographic symmetry analysis suggested that LysIME-EF1 CBD

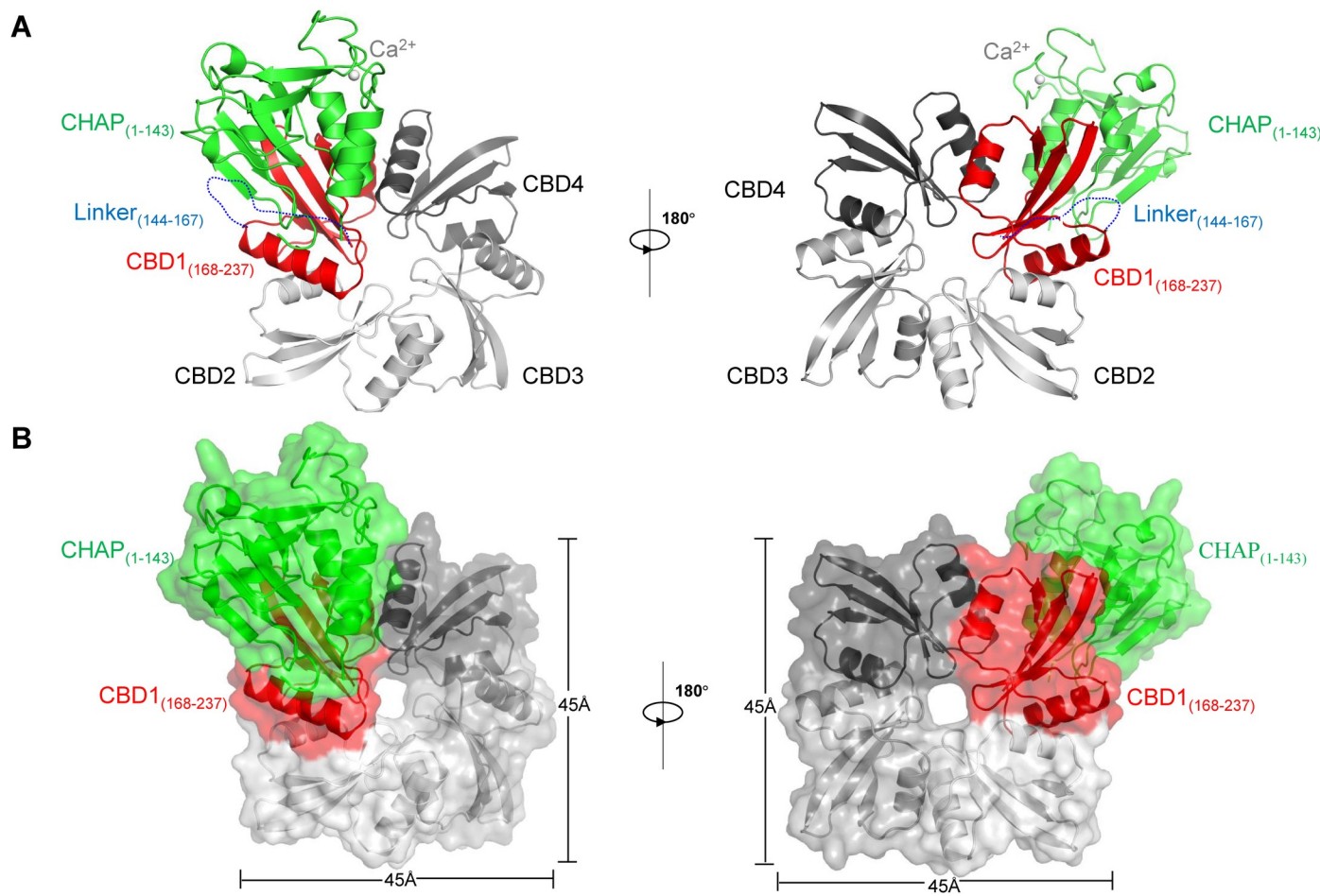

**Fig 2. Overall structure of wild type LysIME-EF1.** **(A)** Ribbon diagram of an asymmetric unit of LysIME-EF1. The full-length LysIME-EF1 was colored in green for CHAP domain and in red for CBD, and the other three CBDs were colored different shades of grey. **(B)** Surface representations of LysIME-EF1 shown the same views and colors as in (**A**). The four CBDs form a tetrameric ring with 45 Å in length and 45 Å in width. The N-terminal CHAP domain is positioned entirely above the CBD tetrameric ring.

forms a stable homotetramer (**Fig 4A**). The oligomeric interface contains 1318.9 Å$^2$ of the buried surface area, accounting for 28.25% of the monomer surface area calculated by PISA [37]. To investigate the oligomerization state of LysIME-EF1 CBD, size-exclusion chromatography was performed. The LysIME-EF1 CBD was cloned into pET 21a(+) with an N-terminal His tag and the purified LysIME-EF1 CBD was subjected to size-exclusion chromatography and the elution profile is 17.7 ml in a Superdex 200 column (GE Healthcare) with an estimated ~32 kDa (**S6A Fig**), supporting that LysIME-EF1 CBD forms a stable tetramer in solution.

Detailed analysis shows that the tetramerization is mainly mediated by extensive hydrogen bonds and hydrophobic interactions (**Fig 4B and 4C**). The hydrogen bonds are mainly formed between the residues located at the α1 helix of one protomer and the residues located at the β4 of the adjacent protomer. More specifically, the residue Y218 forms hydrogen bonds with R208 and Y209 of the adjacent protomer, the carboxyl of R208 and N212, K189 and A231, E201 and N222 also form hydrogen bonds between two adjacent protomers (**Fig 4B**). In addition, a sizeable hydrophobic interface is formed by the neighboring protomers, where the residues M227, M228, A230, A231, L232 of α2 helix from one protomer pack against hydrophobic

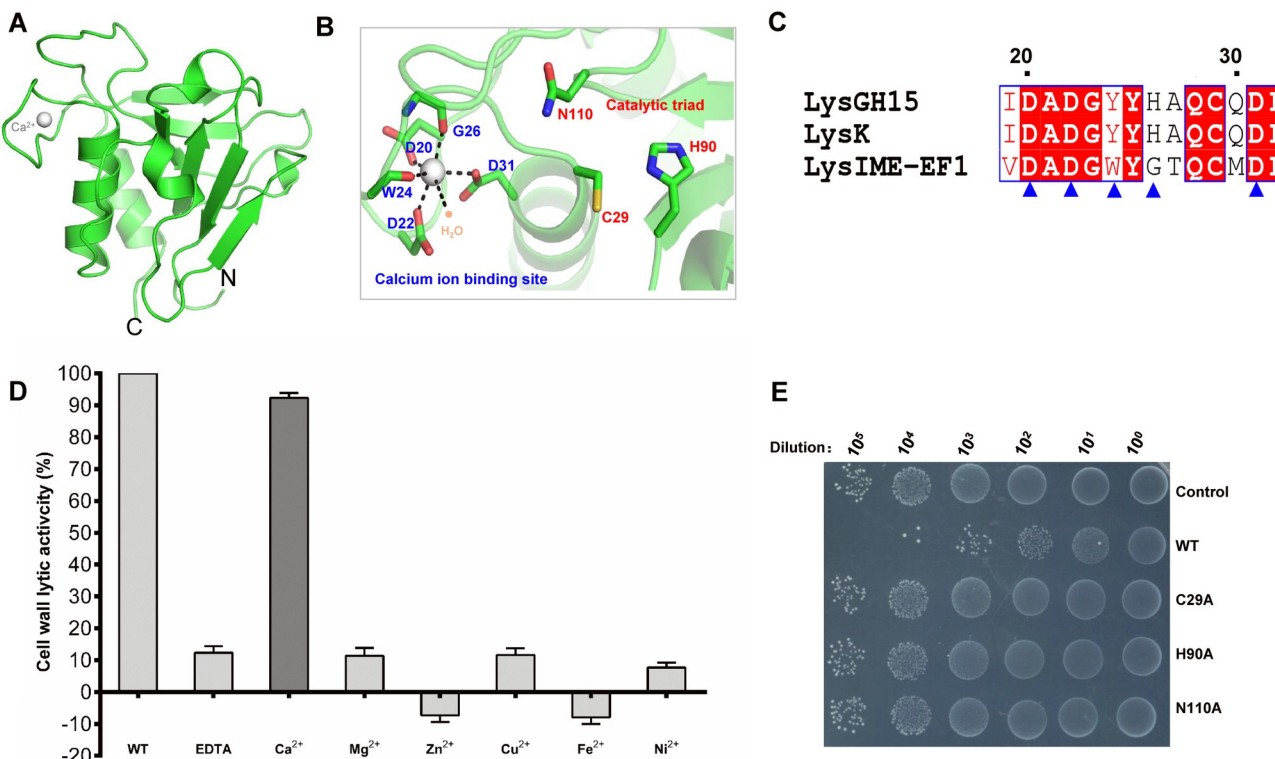

**Fig 3. Structural and functional analysis of the LysIME-EF1 CHAP domain. (A)** Overall structure of the CHAP domain (residues 1–143). The N-termini and C-termini are labeled with the respective letters. The $Ca^{2+}$ ion bound to the LysIME-EF1 CHAP domain is shown as sphere in grey. **(B)** A detailed view of the catalytic triad and the calcium-binding site in the CHAP domain. **(C)** Sequence alignment of the residues involved in calcium binding, which are conserved when aligned with LysGH15 and LysK. The blue triangles indicate the residues involved in $Ca^{2+}$ ion binding. **(D)** The effects of different metal ions (1 µM) on the lytic activity of the EDTA-inactivated WT LysIME-EF1 protein (5 µM) (excess EDTA was removed by dialysis). The values represent the mean ± SD (n = 3). **(E)** Survival test of *E. faecalis* 002 on LB agar dishes after the cells were lysed by the catalytic triad residue (C29A, H90A and N110A) mutants of LysIME-EF1 (0.1 µM) at 37˚C for 1 hours.

patches $_{183}$WFVIGG$_{188}$ and $_{191}$IYLP$_{194}$ of the other protomer. Notably, the M227 nearly inserts into the hydrophobic cavity formed by residues F184, I186, I191 and L193 (**Fig 4C**). In order to evaluate the importance of these residues for tetramerization, we created LysIME-EF1 CBD variants carrying mutants of the corresponding residues. The double mutant that carries an Arg instead of Phe184 and Ala instead of Tyr218 was tested by size-exclusion chromatography. Differing from that of LysIME-EF1 CBD, the elution profile of the LysIME-EF1 CBD F184R&Y218A is 18.7 ml (**S6A and S6B Fig**), confirming that the residues F184 and Y218 are essential for maintaining the oligomeric tetramerization of LysIME-EF1.

## Additional CBD fragments are required for the lytic activity of LysIME-EF1

To further investigate if the three truncated CBD fragments are necessary for the functional LysIME-EF1 holoenzyme, we compared the lytic activities of WT LysIME-EF1 and LysI-ME-EF1 obligate-monomer (LysIME-EF1 F184R/Y218A mutant). Antibacterial activity was assessed as the decrease in viable bacterial counts (**Fig 4D**). The results showed that the antibacterial activity was nearly lost when the interactions between the CBDs were disrupted, suggesting that tetramerization of LysIME-EF1 is required for its highly efficient lysis of *E. faecalis* cells.

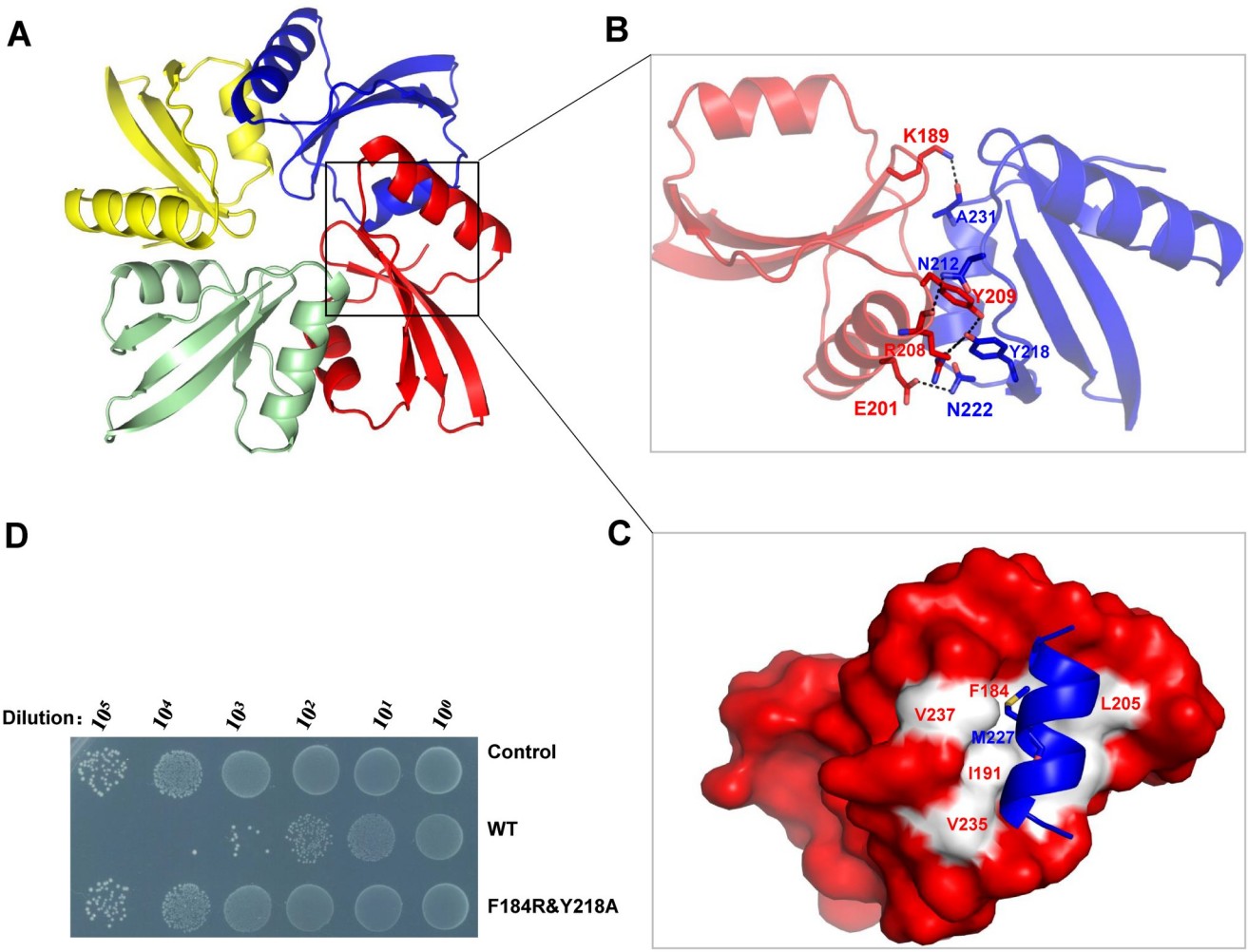

**Fig 4. Analysis of the oligomerization interface of LysIME-EF1 CBD. (A)** Ribbon representation of LysIME-EF1 CBD tetramer. The four monomers (Chains A to D) are shown in red, blue, yellow and palegreen, respectively. **(B)** Ribbon representation showing the detailed interaction between the adjacent CBDs. The key residues are shown in sticks. **(C)** Combined surface representation and a ribbon-stick model showing the molecular interface of the hydrophobic interactions, one LysIME-EF1 CBD is shown in the surface model and the other is shown in the ribbon-sticks model. The hydrophobic residues of LysIME-EF1 in the model are shown in white. **(D)** Survival test of *E. faecalis* 002 on LB agar dishes after the cells were lysed by WT LysIME-EF1 (0.1 μM) as well as LysIME-EF1 F184R/Y218A mutant (0.1 μM) at 37°C for 2 hours.

### Identification of an internal translation site within the *LysIME-EF1* gene

The high-resolution structures of WT LysIEM-EF1 and LysIME-EF1 CBD raised the question about how the extra CBD was produced. To this end, we performed a thorough analysis of the amino acid sequence of LysIME-EF1. Notably, the presence of a methionine (M168) at the N-terminus of the three CBDs in our protein structure prompted us to consider the hypothesis that the site of M168 acts as an alternative in-frame start codon site for translation ($_{502}ATG_{504}$) (**S7 Fig**). Coincidentally, inspection of LysIME-EF1 nucleotide sequence revealed a putative ribosome binding site (RBS) upstream of the alternative start codon (**Fig 5A**).

To establish the relationship between these sequences and the production of truncated CBDs, a methionine-to-leucine substitution (M168L) and mutation of the putative ribosome binding site AA<u>GG</u>AGA to AA<u>CC</u>AGA (mutated nucleotides are underlined) were carried out (hereafter the double mutant protein is designated mLysIME-EF1). Unlike the WT

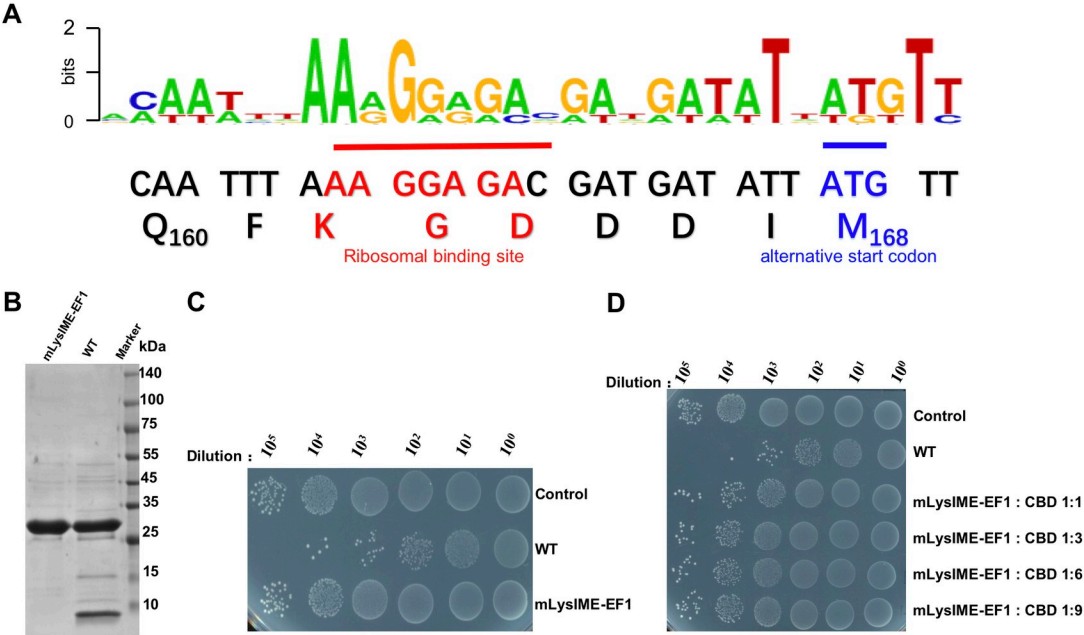

**Fig 5. Identification of an internal ribosomal binding site and alternative start codon within *LysIME-EF1* gene. (A)** Sequence logo created with Weblogo [51] showing the linker region of *LysIME-EF1* gene with nucleotides numbered according to its nucleotide sequence and based on a sequence alignment of LysIME-EF1-related endolysins. The internal ribosomal binding site sequence is underlined with red line, and the start codon is underlined with a blue line. **(B)** SDS-PAGE analysis showing products of the expression of WT LysIME-EF1 and mLysIME-EF1. **(C)** Survival test of *E. faecalis* 002 on LB agar dishes after the cells were lysed by WT LysIME-EF1 (0.1 μM) as well as mLysIME-EF1 (0.1 μM) at 37°C for 2 hours. **(D)** Survival test of *E. faecalis* 002 on LB agar dishes after the cells were lysed by mLysIME-EF1 (0.1 μM) that mixed with CBD at different molar ratio at 37°C for 2 hours.

LysIME-EF1, mLysIME-EF1 is translated into a single product with the molecular mass of ~30 kDa, corresponding to the full-length LysIME-EF1, whereas the ~ 8 kDa product was absent (**Fig 5B**). These results clearly confirmed that both M168 and putative RBS are associated with the production of the ~ 8 kDa CBD fragments. Intriguingly, a similar phenomenon, in which one full-length protein associates with three CBDs produced by secondary translation, has been reported for endolysin Lys170 derived from another *E. faecalis* bacteriophage [32].

In order to further examine the effect of the CBD fragments on lytic activity of full-length LysIME-EF1 against *E. faecalis*, we compared bactericidal potentials of the mLysIME-EF1 with that of the WT LysIME-EF1. As shown in **Fig 5C,** mLysIME-EF1 has significantly reduced bactericidal activity in comparison to the WT LysIME-EF1 protein. This suggests that the co-assembly with three CBDs is a prerequisite for WT LysIME-EF1 to form a functional unit to fight against *E. faecalis*. However, different from Lys170 and CTP1L [32, 34], mixing mLysIME-EF1 with the truncated LysIME-EF1 CBD (residues 168–237) at different molar ratios also could not significantly improve the lytic activity (**Fig 5D**).

The results of the current study and previous reports made us wonder whether the mechanisms of secondary translation and oligomerization are common to other endolysins homologous to LysIME-EF1 and what extent [32]. We performed a protein BLAST using the LysIME-EF1 CBD domain as a query sequence. As a result, 36 unique protein sequences with an E value < 0.01 were identified (**S8 Fig**). All these proteins are described as bacteriophage-derived endolysins that target *Enterococcus* species. Notably, methionine as the first amino acid residue of their CBD domains is a feature common to all of them. In addition, nucleotide sequence alignment revealed the same ribosome binding site (AAGGAGA) sequence present

in all 36 proteins just upstream of the alternative start codon site. These results are consistent with the previous bioinformatics analysis by Proenca et al [32] and imply that the use of a secondary translation initiation site may be widespread among endolysins targeting *Enterococcus* species.

### Identification of the key residues involved in binding to the host cell

Next, for further understanding of the structural basis of how LysIME-EF1 binds to the *Enterococcus* species strains, we aimed to identify the key residues involved in binding to the host cell. A positively charged patch that may contain residues involved in cell-wall binding is located on the planar surface of the LysIME-EF1 tetramer (**S9 Fig**). In order to verify the involvement of this positively charged patch in cell-wall binding, we substituted the residues K173, K176, and R190 with glutamic acid (E) (**Fig 6A**). Then, we generated recombinant N-terminally GFP-tagged LysIME-EF1 CBD variants and assessed their cell-wall binding capability using fluorescence-activated cell sorting (FACS) (**Fig 6B**). When compared to the WT LysIME-EF1, mutations K173E and K176E had a slight impact on LysIME-EF1 cell-wall binding ability, whereas the mutation R190E nearly abolished the cell-wall binding ability of LysIME-EF1. In addition, we also compared bactericidal activities of the abovementioned LysIME-EF1 variants against *E. faecalis* 002. As expected, the R190E mutant exhibited almost no lytic activity (**Fig 6C**), which may be due to substantial loss of host cell binding ability. These

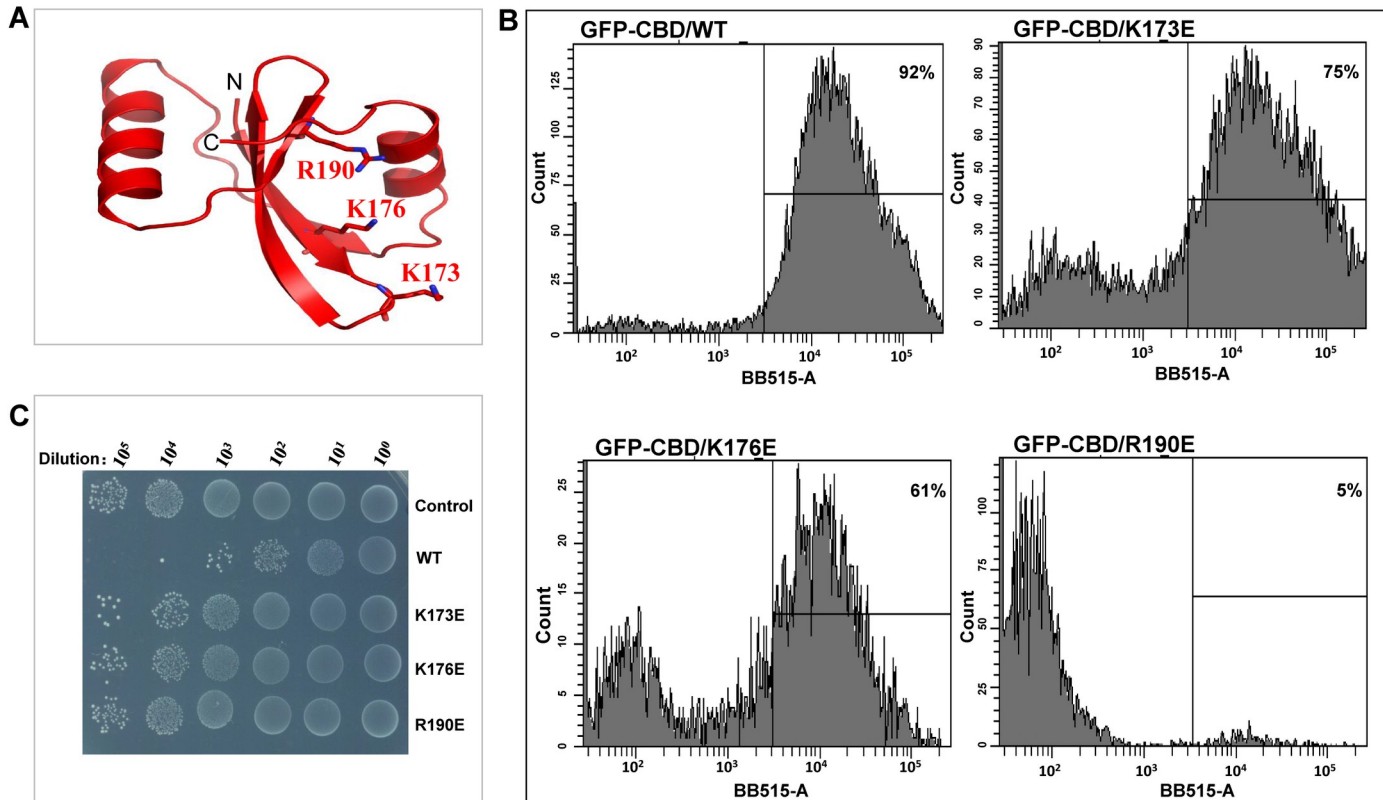

**Fig 6. Key residues of LysIME-EF1 CBD involved in binding to *E. faecalis* cell-wall.** (A) Cartoon depiction of LysIME-EF1 monomer shows the predicted residues (K173, K176 and R190) involved in cell-wall binding in sticks. (B) Flow cytometry was used to assess the ability of LysIME-EF1 and its variants (WT LysIME-EF1 and K173E, K176E and R190E mutants) to bind *E. faecalis* 002. (C) Survival test of *E. faecalis* 002 on LB agar dishes after the cells were lysed by WT LysIME-EF1 (0.1 μM) as well as K173E, K176E and R190E mutants (0.1 μM) at 37˚C for 2 hours.

results suggest that R190 is the key residue mediating binding of LysIME-EF1 to host cells, which is in accordance with its high level of conservation in LysIME-EF1 and homologous endolysins (**S8 Fig**).

## Evaluation of protective capabilities in mice

Finally, to investigate whether the additional CBDs and the residues involved in cell-wall binding are necessary for protection against *E. faecalis* challenge in mice *in vivo*, the purified WT LysIME-EF1 and its derivatives (mLysIME-EF1, LysIME-EF1 C29A/H90A/N110A, K173E, K176E, R190E, or F184R/Y218A mutants) were administered to mice inoculated intraperitoneally with *E. faecalis* 002. When determining the minimal lethal dose (MLD) of *E. faecalis* 002, we found that injecting $2\times10^9$ CFU (colony-forming unit)/mouse of *E. faecalis* 002 caused nearly a 100% mortality rate within 4 days. The dose of 2×MLD (i.e. $4\times10^9$ CFU/mouse) was then used as the challenge dose for testing the protection capabilities of WT LysIME-EF1 and its derivatives. Equal amounts of WT LysIME-EF1 (20 μg) or LysIME-EF1 derivatives (20 μg) were administered to respective experimental groups (n = 24) 1 h post challenge inoculation. The results revealed that the mice administered WT LysIME-EF1 provided a 95.83% (23 out of 24) protection rate. However, the catalytic triad triple mutant (C29A/H190A/N110A) provided only a 12.5% protection rate and mLysIME-EF1 provided a 25% protection rate, whereas the LysIME-EF1 F184R/Y218A rescued only 16.67% of mice. The protection rates of K173E, K176E, R190E LysIME-EF1 were 54.17%, 45.83%, 16.67%, respectively (**Fig 7**). Taken together, the animal experiments further confirmed that the additional LysIME-EF1 CBD and the residues involved in cell-wall binding on CBD are important for the lytic activity of LysIME-EF1.

## Discussion

Use of bacteriophage-derived endolysins is nowadays considered a potential strategy for treatment of bacterial infections [38–40], particularly caused by multidrug-resistant pathogenic microorganisms [41]. Despite its potent bactericidal effect and potential for therapeutic application, limited structural and functional information of endolysins has been available. The high-resolution X-ray structures of full-length LysIME-EF1 and its CBD fragments reported in this study contribute to our understanding of the multimeric endolysins and provide a template for elucidating the biological function and underlying mechanisms of LysIME-EF1 and its highly efficient lytic activity.

The structure of WT LysIME-EF1, which is distinct among known structures of endolysins, consists of one full-length protein associated with three additional N-terminally truncated CBD peptides. The structure of the CBD in the full-length LysIME-EF1 protein does not show any differences from that of the truncated CBD peptides. However, disruption of the oligomerization interface between CBDs results in significant decrease in lytic activity against *E. faecalis*, suggesting that LysIME-EF1 is a functional endolysin that requires the assembly of additional three CBDs to achieve an oligomeric conformation for optimal activity. A similar result was reported in other multimeric endolysins, such as the previously identified endolysin Lys170 from the enterococcal phage F170/08 or CTP1L, which lyses *Clostridium tyrobutyricum* [32, 34]. However, the molecular mechanism of the multimeric endolysins recognition of their substrates remains to be illuminated.

The presence of secondary translation products has been described for endolysins of several phages [42–44], which is consistent with the generally accepted notion that bacteriophages make very economical use of their genetic material. For instance, in-frame secondary translation site has previously been reported for endolysin of the staphylococcal phage 2638A, which is lytic for *Staphylococcus aureus* [42]. Concerning oligomerization, only three endolysins have

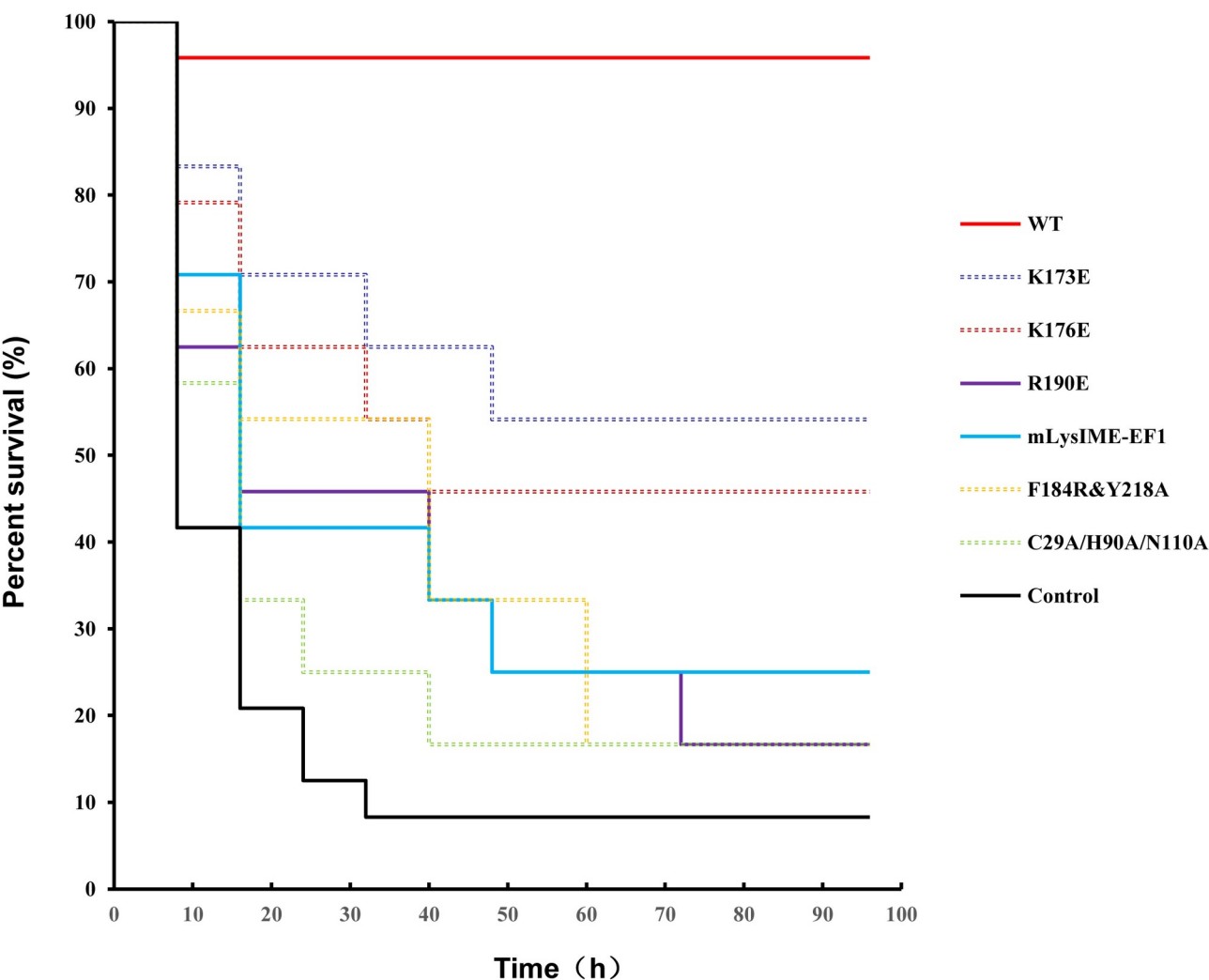

**Fig 7. LysIME-EF1 rescues mice from *E. faecalis* infection.** Mice were injected intraperitoneally with $4\times10^9$ CFU/mouse of *E. faecalis* 002. One hour later they were treated with WT LysIME-EF1 and LysIME-EF1 variants proteins (20 μg in 200 μl 1xPBS buffer) by the same route and followed for 4 days (n = 24). As a blank control, mice were treated with an equal volume of 1xPBS buffer. Mice survival rate was evaluated using Kaplan–Meier analysis. For clarity, overlapping curve lines are shown as dotted lines.

been reported to form an active multimer: CTP1L and CD27L from *Clostridium tyrobutyricum* phages, and Lys170 from *E. faecalis* phage [32–34]. The results of our study and the abovementioned reports suggest that the use of an internal translation site may be common among endolysin genes. Structure-based analysis of the LysIME-EF1 CBD domain and homology search identified 36 endolysins from *E. faecalis* phages, which display amino acid sequence identity ranging from 37% to 98% with LysIME-EF1. The common feature of these endolysins is the presence of an internal ribosomal binding site upstream of a putative alternative start codon.

Based on the results of the structural and biochemical analysis, it was demonstrated that the residue R190 is essential for LysIME-EF1 binding to host cell. The animal experiments performed in this study further corroborated the results of structural and biochemical analysis.

In conclusion, our work sheds light on the molecular mechanism by which WT LysIME-EF1 lyses host cells, and thus lays the basis for designing endolysins with high activity and broader antimicrobial spectrum that can be useful for pharmaceutical applications.

## Materials and methods

### Ethics statement

Animal use protocols are approved by the Animal Ethics and Welfare Committee of Beijing Institute of Microbiology and Epidemiology (#2018–583) and implemented in accordance with the national standards of the People's Republic of China (GB / T 35892–2018: Guidelines for the review of Experimental mice-animal welfare ethics, GB / T 35823–2018: General requirements for laboratory animal-animal experiments).

### Protein expression and purification

The coding sequence of the LysIME-EF1 (NC_041959.1) was PCR-amplified using primers (forward: 5'-gcggatccatggttaaattaaacgatgtac-3', reverse: 5'- gctctagactatactttaacttgtggtaaagc-3'), after which the amplified fragment was cloned into the expression vector pCold I using restriction endonuclease *Xba* I and *Bam*H I (NEB). The construct was introduced into *E. coli* BL21 (DE3) cells for expression of recombinant proteins. The recombinant strains were cultured in LB medium (10 g/L NaCl, 10 g/L tryptone, and 5 g/L yeast extract) at 37°C until the $OD_{600}$ reached 0.8. After 16 h of induction with 0.5 mM isopropylthio-beta-D-galactoside (IPTG) at 16°C, the cells were harvested by centrifugation at 2,000×g for 30 min at 4°C. The pellet was resuspended in the lysis buffer containing 50 mM Tris-HCl (pH 8.0) and 150 mM NaCl, and lysed by ultrasonication. The lysate was then centrifuged at 23,000×g for 30 min at 4°C, and the supernatant was loaded onto a Ni-NTA column (Qiagen). The target protein was eluted in buffer containing 50 mM Tris-HCl (pH 8.0), 150 mM NaCl and 250 mM imidazole. After sample analysis with SDS-PAGE, the fractions containing pure protein were pooled and concentrated to 0.5 ml, and then loaded onto a Superdex 200 increase column (GE Healthcare) equilibrated with a buffer containing 20 mM Tris-HCl (pH 8.0) and 150 mM NaCl. Seleno-methionine derivate of LysIME-EF1 was purified using the same procedure, but with 2 mM DTT added to the buffer. All mutants of LysIME-EF1 were purified in the same procedure as the WT LysIME-EF1.

The CHAP domain (residues 1–143) and the CBD (residues 168–237) of LysIME-EF1 were sub-cloned into pET21a with an N-terminal 6xHis-tag, the recombinant protein was expressed and purified as the WT LysIME-EF1.

### Site-directed mutagenesis

The Quick-Change Site-Directed Mutagenesis Kit (Stratagene) was used to generate LysIME-EF1 mutants. The primers used to generate the mutants are listed in **S1 Table**. All mutants were constructed and verified by sequencing of the entire gene. Expression and purification of the mutants were performed using the same method as the production of the WT LysIME-EF1.

### Crystallization

Initial crystallization screens were conducted with sitting-drop vapor diffusion method using commercial crystallization kits. The protein concentration used for crystallization was 15 mg/ml. Hampton Research kits were used in the sitting drop vapor diffusion method to identify preliminary crystallization conditions at 16°C. Crystallization drops contained 0.5 μl of the protein solution mixed with 0.5 μl of reservoir solution. Crystals appeared in two conditions. Cubic-shaped crystals appeared in 0.1 M HEPES (pH 6.5–7.5), 50 mM magnesium acetate and 20% (w/v) PEG 4,000. Diamond-shaped crystals appeared in 20% PEG8,000 and 0.2 M sodium potassium tartrate tetrahydrate 0.1 M Tris pH 8.5. Crystals were harvested with 20% (v/v)

ethylene glycol as cryoprotectant before flash freezing in liquid nitrogen. To solve phase problem, selenomethionine was introduced with the standard method. The protein concentration of the selenomethionine derivate of LysIME-EF1 used for crystallization was 30 mg/ml. Crystals of LysIME-EF1$^{SeMet}$ were grown in 0.1 M MES (pH 5.6–6.2), 0.2 M Ca(OAc)$_2$, and 10–14% (v/v) 2-propanol. All crystals were flash-frozen in liquid nitrogen, with the addition of 20%-25% (v/v) glycerol as cryo-protectant.

## Data collection and structure determination

X-ray diffraction data were collected at the beamline BL-17U1 of the Shanghai Synchrotron Radiation Facility (SSRF). All data were processed using HKL-2000 [45, 46]. The original structure of LysIME-EF1 was determined by using the single-wavelength anomalous dispersion (SAD) phasing. Phases were calculated using AutoSol implemented in PHENIX [47]. AutoBuild in PHENIX was used to automatically build the atom model. The structure of the CBD of LysIME-EF1 was determined by using molecular replacement with Phaser [48]. After several rounds of positional and B-factor refinement using Phenix, as well as refinement with TLS parameters alternated with manual model revision using Coot [49], the quality of final models were checked using the PROCHECK program. Details of the data collection and refinement statistics are given in **Table 1.** All of the figures showing protein structures were prepared with PyMOL [36]. All coordinates and structure factor files are available at the PDB database as 6IST (WT LysIME-EF1) and 6L00 (LysIME-EF1 CBD).

## Size-exclusion chromatography

The purified WT LysIME-EF1, mLysIME-EF1, LysIME-EF1 CHAP (residues 1–143), LysIME-EF1 CBD (residues 168–237), LysIME-EF1 F184R/Y218A and LysIME-EF1 CBD F184R/Y218A were subjected to size-exclusion chromatography (Superdex 200 10/300 GL column, GE Healthcare, 20 mM Tris–HCl pH 8.0, 150 mM NaCl), respectively. The assay was performed at a flow rate of 0.5 ml/min at 16˚C. Standard proteins containing Thyroglobulin (669 kDa), Aldolase (158 kDa), Conalbumin (75 kDa), Ovalbumin (44 kDa), Aprotinin (6.5 kDa) were performed in the same buffer at the same condition.

## Analytic ultracentrifugation (AUC)

Sedimentation velocity experiments were used to assess the molecular sizes of the WT of LysIME-EF1 and the CBD of LysIME-EF1 at 20˚C on a Beckman XL-A analytical ultracentrifuge equipped with absorbance optics and an An60 Ti rotor (Beckman Coulter, Inc. Fullerton, CA). Samples were diluted to an optical density at 280 nm (OD$_{280}$) of 1 in a 1.2 cm path length. The rotor speed was set to 35,000×g for all samples. The sedimentation coefficient was obtained using the c(s) method with the Sedfit Software [50].

## Turbidity reduction assay

*E. faecalis* 002 *was* grown to mid-exponential growth phase until OD$_{600}$ reached 0.8, after which the bacteria were centrifuged at the speed of 3,000×g for 10 mins. The cells were washed three times with 1xPBS buffer and then resuspended in 3 ml of fresh LB medium. Eighteen microliters of the purified endolysin protein (1 mg/ml) were added to the bacterial sample, and the mixture was cultured on a shaking platform at 37˚C. Changes in value of OD$_{600}$ were measured every 20 min.

EDTA (1 mM) was added to the purified wild-type LysIME-EF1 and the excess EDTA was removed by dialysis using 1xPBS buffer at 4˚C for 16 h. Afterwards, equal amounts of different

metal ions (1 μM) were added to the EDTA-inactivated LysIME-EF1 and incubated for 2 h at 4˚C. Peptidoglycan lytic activity was measured as the decrease in the $OD_{600}$.

## Bactericidal activity assay

For the antimicrobial activity assay, equimolar concentrations (0.1 μM) of LysIME-EF1 and LysIME-EF1 mutants were added into *E. faecalis* 002 bacterial suspensions ($OD_{600}$ = 0.8), respectively. After 2 hours of incubation at 37˚C, 220 rpm. The cell suspension was washed twice using 1xPBS buffer by centrifugation (17,000×g, 10 min), and then 4 μl of tenfold serially diluted cell suspension were spot plated onto LB agar dishes for survival assay. Residual viable cell numbers on the plate were measured after incubation at 37˚C for 12 h. For the blank control, the same volume of 1xPBS buffer was added instead of endolysin LysIME-EF1. The antibacterial activity was expressed as the decrease in viable bacterial counts. All experiments were performed in triplicate.

To test the effect of the CBDs on the wild-type LysIME-EF1 and the mLysIME-EF1, wild-type LysIME-EF1 or mLysIME-EF1 were incubated with LysIME-EF1 CBD at different molar ratios (1:1, 1:3, 1:6, 1:9) at 4˚C for 2 h. The method of measuring bactericidal activity is described as above.

## Flow cytometry analysis

*E. faecalis* 002 was cultured in LB medium until the $OD_{600}$ reached 0.8, after which the cells were centrifuged at the speed of 3,000×g. The obtained pellet was washed and resuspended in 1xPBS buffer. The purified GFP-LysIME-EF1 CBD was mixed with the resuspended *E. faecalis* 002 and incubated at 4˚C for 1 hour. After two washes with 1xPBS buffer, the percentage of GFP-positive cells was determined using the Cell Quest Software (Becton Dickinson).

## Protection by LysIME-EF1 in mouse infected with *E. faecalis* 002

The infection model was based on methods described previously [35], using 6–8 weeks-old female BALB/c mice (weight range, 18–20 g) obtained from the Laboratory Animal Center of the Academy of Military Medical Sciences, Beijing, China. *E. faecalis* 002 was grown to $OD_{600}$ = 1.6, and the bacteria were centrifuged at 10,000×g, 10 min, 4˚C. The bacterial pellet was resuspended in 1xPBS buffer. Bacterial inoculation titers were calculated by serial dilution and plating onto BHI agar plates for each experiment. Mice were injected with different inocula of the *E. faecalis* 002, i.e. $2×10^7$, $2×10^8$, $2×10^9$ and $2×10^{10}$ CFU (colony forming unit) in 200 μl of 1xPBS buffer to determine the minimal lethal dose (MLD) that would cause 100% mortality over a 4-day period. Once the MLD was determined to be $2×10^9$ CFU, 2xMLD (i.e. $4×10^9$ CFU) was used as the infective inoculum (the challenge dose) in the mouse model. To determine the protection effect of the LysIME-EF1 and its variants in mice challenged with 2xMLD of the *E. faecalis* 002, a single dose (20 μg in 200 μl of 1xPBS buffer) of LysIME-EF1 proteins (WT LysIME-EF1, mLysIME-EF1, LysIME-EF1 C29A/H90A/N110A, K173E, K176E, R190E, and F184R/Y218A mutants) were administered intraperitoneally on the other side of abdomen 1 h post bacterial inoculation. Each experimental group contained 24 mice. In the blank control group, the infected mice were treated with equal volume (200 μl) of 1xPBS buffer. The survival rate for each experimental group was monitored every 8 h during the first 48 h and then every 12 h for up to 4 days post-infection.

## Statistical analysis

Statistical significance was determined using the unpaired two-tailed Student's *t*-test at a level of significance of $P<0.05$. Survival data were analyzed using a Kaplan–Meier survival analysis with a log-rank method.

## Supporting information

**S1 Fig. LysIME-EF1 possesses efficient bactericidal spectrum against 29 clinical strains of**
***E. faecalis.*** A decrease in $OD_{600}$ was used to evaluate the lytic activity of LysIME-EF1. WT
LysIME-EF1 was used to lyse against *E. faecalis*. All assays were performed in triplicate, and
the data are expressed as means ± SD (n = 3).
(TIF)

**S2 Fig. The oligomerization of wild type LysIME-EF1 in solution. (A)** Size-exclusion chro-
matography (GE Healthcare, Superdex 200 increase, buffer: 20 mM Tris-HCl, pH 8.0, 150 mM
NaCl) of WT LysIME-EF1. The CBD fragment of LysIME-EF1 was co-eluted with the full-
length LysIME-EF1 at 15.5 mL, standard proteins containing Thyroglobulin (669 kDa), Aldol-
ase (158 kDa), Conalbumin (75 kDa), Ovalbumin (44 kDa), Aprotinin (6.5 kDa) were per-
formed in the same buffer. **(B)** Analytical ultracentrifugation (AUC) of the exact molecular
mass of WT LysIME-EF1, which yielded a sedimentation coefficient of 3.264 S with a molecu-
lar mass of 55.6 kDa. The c(s) represents continuous (sedimentation coefficient distribution)
analysis model.
(TIF)

**S3 Fig. The size-exclusion chromatography and purification of LysIME-EF-1 CHAP and
CBD. (A)** Size-exclusion chromatography (GE Healthcare, Superdex 200 increase, buffer: 20
mM Tris-HCl, pH 8.0, 150 mM NaCl) of LysIME-EF1 CHAP, standard proteins containing
Thyroglobulin (669 kDa), Aldolase (158 kDa), Conalbumin (75 kDa), Ovalbumin (44 kDa),
Aprotinin (6.5 kDa) were performed in the same buffer. **(B)** The SDS-PAGE analysis of LysI-
ME-EF1 CHAP and CBD.
(TIF)

**S4 Fig. LysIME-EF1 possesses a classical CHAP. (A)** The structure of the CHAP domain of
LysIME-EF1. **(B)** The superposition of the CHAP domains of LysIME-EF1, LysK and
LysGH15. **(C)** Structure-based sequence alignment of LysIME-EF1, LysGH15 and LysK
CHAP domain. Strictly conserved residues are boxed in white on a red background and highly
conserved residues are boxed in red on a white background. At the top of the sequences, the
secondary structure elements of LysIME-EF1 CHAP are shown, and every 10 residues are
indicated with a dot (·) shown above the sequences. Sequence alignment was generated by
ClustalW (https://www.genome.jp/tools-bin/clustalw) and the figure was generated by ESpript
3 (http://espript.ibcp.fr/ESPript/ESPript/). The blue triangles indicate the residues involved in
$Ca^{2+}$ ion binding and the red triangles indicate the catalytic triad residues on CHAP domain.
(TIF)

**S5 Fig. Crystal structure of LysIME-EF1 CBD. (A)** Overview of LysIME-EF1 CBD monomer
with labeled secondary elements. The LysIME-EF1 CBD monomer was shown in red color.
**(B)** Topological diagram of LysIME-EF1 CBD with β-strands drawn in black arrows and the
α-helices shown as cylinders.
(TIF)

**S6 Fig. LysIME-EF1 CBD is a homotetramer in solution. (A)** Size-exclusion chromatogra-
phy (GE Healthcare, Superdex 200 increase, buffer: 20 mM Tris-HCl, pH 8.0, 150 mM NaCl)
of LysIME-EF1 CBD and the mutant LysIME-EF1 CBD F184R/Y218A. The elution peaks for
the two proteins were 17.7 ml and 18.7 ml, respectively. **(B)** The SDS-PAGE analysis of LysI-
ME-EF1 CBD F184R&Y218A protein.
(TIF)

**S7 Fig. The N-terminal residue identification of the LysIME-EF1 CBD.** A blue colored monomer extracted from the structure of Fig 4A. The first four residues of LysIME-EF1 CBD including $M_{168}F_{169}I_{170}Y_{171}$ are shown in sticks and the 2Fo-Fc map was contoured at 1.0 σ level.
(TIF)

**S8 Fig. Sequence alignment of LysIME-EF1 CBD with CBD of other endolysins derived from *E. faecalis* phages.** Multiple sequence alignment of LysIME-EF1 CBD and CBD of other endolysins of *E. faecalis* phages produced by ClusterW (https://www.genome.jp/tools-bin/clustalw) and ESpript 3 (http://espript.ibcp.fr/ESPript/ESPript/). Every 10 residues are indicated with a dot (·) shown above the sequences. Strictly conserved residues are indicated in white on a red background. The blue triangles indicate the positive residues involved in putative cell-wall binding and the red triangles indicate the residues responsible for the oligomerization of the CBD.
(TIF)

**S9 Fig. The electrostatic surface of LysIME-EF1.** The positively charged patch is located on the planar surface of the LysIME-EF1 CBD tetramer. The putative cell binding residues K173, K1746 and R176 are shown in sticks. Positively and negatively charged regions are colored in blue and red, respectively.
(TIF)

**S1 Table. Primers used to generate the LysIME-EF1 mutants in this study.**
(TIF)

## Acknowledgments

The authors thank the staff at beamline BL17-U1 of Shanghai Synchrotron Radiation Facility for their help with the X-ray diffraction data collection.

## Author Contributions

**Conceptualization:** Yigang Tong, Songying Ouyang.

**Data curation:** Biao Zhou.

**Funding acquisition:** Xiangkai Zhen, Zhiqiang Mi, Songying Ouyang.

**Investigation:** Biao Zhou, Xiangkai Zhen, Huan Zhou, Feiyang Zhao, Chenpeng Fan.

**Project administration:** Yigang Tong, Songying Ouyang.

**Software:** Biao Zhou, Huan Zhou.

**Writing – original draft:** Xiangkai Zhen, Vanja Perčulija.

**Writing – review & editing:** Songying Ouyang.

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
