## [Decision Letter · Decision Letter 0]

24 Oct 2019

Dear Prof. Ouyang,

Thank you very much for submitting your manuscript "Structural and functional insights into a novel two-component endolysin encoded by a single gene in Enterococcus faecalis phage" (PPATHOGENS-D-19-01755) for review by PLOS Pathogens. Your manuscript was fully evaluated at the editorial level and by independent peer reviewers. The reviewers appreciated the attention to an important problem, but raised some substantial concerns about the manuscript as it currently stands. These issues must be addressed before we would be willing to consider a revised version of your study. We cannot, of course, promise publication at that time.

We therefore ask you to modify the manuscript according to the review recommendations before we can consider your manuscript for acceptance. Your revisions should address the specific points made by each reviewer.

(1) A letter containing a detailed list of your responses to the review comments and a description of the changes you have made in the manuscript. Please note while forming your response, if your article is accepted, you may have the opportunity to make the peer review history publicly available. The record will include editor decision letters (with reviews) and your responses to reviewer comments. If eligible, we will contact you to opt in or out.

(2) Two versions of the manuscript: one with either highlights or tracked changes denoting where the text has been changed; the other a clean version (uploaded as the manuscript file).

Additionally, to enhance the reproducibility of your results, PLOS recommends that you deposit your laboratory protocols in protocols.io, where a protocol can be assigned its own identifier (DOI) such that it can be cited independently in the future. For instructions see http://journals.plos.org/plospathogens/s/submission-guidelines#loc-materials-and-methods

We hope to receive your revised manuscript within 60 days. If you anticipate any delay in its return, we ask that you let us know the expected resubmission date by replying to this email. Revised manuscripts received beyond 60 days may require evaluation and peer review similar to that applied to newly submitted manuscripts.

There is additional documentation related to this decision letter. To access the file(s), please click the link below. You may also login to the system and click the 'View Attachments' link in the Action column.

[LINK]

Editor added:  It is essential to address all questions raised by the third reviewer for revised version.

Sincerely,

Gongyi Zhang

Associate Editor

PLOS Pathogens

Michael Otto

Section Editor

PLOS Pathogens

Kasturi Haldar

Editor-in-Chief

PLOS Pathogens

orcid.org/0000-0001-5065-158X

Grant McFadden

Editor-in-Chief

PLOS Pathogens

orcid.org/0000-0002-2556-3526

Reviewer's Responses to Questions

**Part I - Summary**

Reviewer #1: Zhou and coworkers report the first structure of a known E. faecalis-targeted endolysin, LysIME-EF1 and find it posing multimeric composition, i.e., a full-length endolysin with three extra CBDs. A lysin (Lys170, the endolysin from enterococcal phage F170/08) with similar composition has been reported by reference 28 (Proenca D, Velours C, Leandro C, Garcia M, Pimentel M, et al. A two-component, multimeric endolysin encoded by a single gene. Mol Microbiol 2015，95: 739-753). The difference is that the current finding is based on the crystallographic study of LysIME-EF1, which is important for further understanding the action model of similar lysins against E. faecalis.

Reviewer #2: Lysins are hydrolytic enzymes produced by bacteriophages in order to cleave the host’s cell wall and are considered as a potential strategy for treatment of bacterial infections caused by multidrug-resistant pathogenic microorganisms. This study focuses on LysIME-EF1, an endolysin that specifically lyses Enterococcus faecalis, a major pathogen with high levels of antibiotic resistance.

In this study, the authors observed that expression of LysIME-EF1 results in two polypeptides, a full length product and a C-terminal CBD domain. Both polypeptides are necessary for the assembly of the functional lysin. They further determined the crystal structure of LysIME-EF1 at a high resolution. The structure revealed an asymmetric assembly containing one full length LysIME-EF1 and three CBD domains. Based on the structure, the authors identified potential key residues, designed mutants and assessed their cell binding capability. Overall, the study constitutes a novel insight into this therapeutically important enzyme and open the possibilities for the future structure-based design of lysins with higher activity and broader antimicrobial spectrum.

Reviewer #3: See attached review

**Part II – Major Issues: Key Experiments Required for Acceptance**

Reviewer #1: 1. Fig 1B and Fig 5B. The images show that the band intensity of the CBD (8 kD) is much lighter compared with that of the full length 30 kD LysIME-EF1. How to explain this result if it is correct that the purified LysIME-EF1 consists of a full-length of 30 kD LysIME-EF1 and three extra 8 kD of CBDs? In theory, the concentration of the CBD should be 3 times of that of the full length 30 kD LysIME-EF1 during SDS-PAGE electrophoresis. For instance, both bands of Lys170 in the SDS-PAGE showed similar density in the previous report (Ref. 20).

2. Fig 5C: Additional experiments should be done by mixing M168L/RBS LysIME-EF1 mutant with different ratio of the purified CBD to see if the lytic activity could be restored in order to confirm the multimeric composition.

3. Fig 1C, 3E and 6C: From the lysis curves of the mutants, it is hard to judge the activity difference between them. It is suggested to use the more sensitive culture methods, such as plate counting, to confirm the activity of different mutants, especially K173E, K176E, C29A, H90A, N110A, CHAP, CBD, and CHAP+CBD.

4. Fig 7 and all related text: The in-vivo infection model might just prove that the protection rates are related to the activity of the lysin used, which did not add more information to the structure of LysIME-EF1 and could be deleted.

Reviewer #2: (No Response)

Reviewer #3: See attached review

**Part III – Minor Issues: Editorial and Data Presentation Modifications**

Reviewer #1: 1. Page 2, lines 27-2: Most natural endolysins has limited host range, but there are several lysins with extended killing spectra, for instance, chimeric lysin ClyR, natural lysin PlySs2 and PlyV12. What does low persistence mean? Please rephase this sentence.

2. Page 4, lines 59: Enterococcus faecalis should be written as E. faecalis after appearing the first time. Correct this throughout the text.

3. Line 10, page 187, Supporting figure is missing.

4. Page 12, line 214：Interaction of lysin with bacterial cells is not infection. Post-infection could be changed to post-addition. The same mistake also goes to page 13, line 239.

5. Page 12, line 216: What does “ability” mean?

6. Page 15, line 277: 10 hours post infection? In the Figure 7 legend and the material method, it is mentioned that lysins were administrated 1 h post infection.

Reviewer #2: Nonetheless, I have the following comments about the manuscript:

1. The author reported that one full length LysIME-EF1 and three CBD domains co-assemble to a functional lysin. Is this the only possible assembly? Does the CHAP domain block the additional CHAP domain to assemble on a CBD tetramer?

2. The authors reported that the CHAP alone, CBD alone or a mixture of the two constructs, exhibited no bactericidal activity. This indicates that 1) CHAP domain and CBD tetramer can not co-assemble in vitro. 2) assembly with the CBD tetramer likely has an allosteric effect on the CHAP domain. The structural comparison of CHAP domain alone and that in complex with CBD will be helpful.

3. The presence of secondary translation products has been observed previously in other proteins, such as MthK (Jiang et al, Nature, 2002) and Tvok (Parfenova et al, J Biol Chem, 2007).

Reviewer #3: See attached review

PLOS authors have the option to publish the peer review history of their article (what does this mean?). If published, this will include your full peer review and any attached files.

Reviewer #1: Yes: Hongping Wei

Reviewer #2: No

Reviewer #3: No

---

## [Decision Letter · Decision Letter 1]

29 Jan 2020

Dear Prof. Ouyang,

Thank you very much for submitting your manuscript "Structural and functional insights into a novel two-component endolysin encoded by a single gene in Enterococcus faecalis phage" for consideration at PLOS Pathogens. As with all papers reviewed by the journal, your manuscript was reviewed by members of the editorial board and by several independent reviewers. The reviewers appreciated the attention to an important topic. Based on the reviews, we are likely to accept this manuscript for publication, providing that you modify the manuscript according to the review recommendations.

Sincerely,

Gongyi Zhang

Associate Editor

PLOS Pathogens

Michael Otto

Section Editor

PLOS Pathogens

Kasturi Haldar

Editor-in-Chief

PLOS Pathogens

orcid.org/0000-0001-5065-158X

Michael Malim

Editor-in-Chief

PLOS Pathogens

orcid.org/0000-0002-7699-2064

Reviewer Comments (if any, and for reference):

Reviewer's Responses to Questions

**Part I - Summary**

Reviewer #1: (No Response)

Reviewer #2: The authors have done a good job at explaining some key points and carrying out additional informative experiments. I believe the paper is ready for publication.

Reviewer #3: I was reviewer 3 in the original manuscript and this revised manuscript is significantly improved. In particular, the authors have added new bacteriolytic assays for most experiments, added additional purification gels, shown standards on gel filtration, and included additional details for some of the methodology. Moreover, they have adequately addressed most (not all) reviewer comments to a reasonable level of satisfaction. Nonetheless, there still remain a couple of errors and questions that were not fully addressed, requiring some additional revisions.

**Part II – Major Issues: Key Experiments Required for Acceptance**

Reviewer #1: (No Response)

Reviewer #2: (No Response)

Reviewer #3: Line numbers in the comments below refer to the marked up version of the revised manuscript.

- Lines 108/109 and 122/123. The authors state that their enzyme “displays a broader host range than other homologous endolysins, with bactericidal activity against 29 clinical strains of E. faecalis”. What do they mean by this comment? A broader host range than what? The fact that they tested 29 strains does not make this broad activity or broad spectrum since all 29 strains were the exact same species. All endolysins show activity toward at least one species and many show activity toward multiple species. Thus, to make a statement about a broader host range than other endolysins when they only test one species is nonsensical.

- Line 209. The authors reference figure S3C, which does not exist. I believe they mean to reference figure S4C.

- Line 225. The authors reference figures S4A and S4B, but I believe they mean to reference figures S5A and S5B.

- Lines 313-325. Proenca et al. 2015 did this same bioinformatics analysis on Lys170 and found the same results (i.e. methionine as first amino acid residue of the CBD and upstream Shine-Dalgarno sequence suggesting an alternative start codon site that generates oligomer). At the least, the authors need to acknowledge this previous work. Perhaps a statement saying that their analysis is consistent with the findings of Proenca et. al or something along these lines.

- Lines 332-336. The authors discuss residues 173, 176, and 190 and label these residues in figure 6A. This is fine; however, it would be very insightful to the reader if they also pointed out the location of these residues in Figure S9, which shows the electrostatic map of the planar surface so the reader can visualize exactly where these critical residues are located on the CBD surface.

- Lines 338-340. The authors should insert a reference to Figure 6B at the end of this sentence describing the FACS results.

- Line 346. The authors reference Figure 6B here, but I believe they mean to reference Figure S8 instead.

- Figure 7. This figure and all of the associated in vivo work is still fraught with issues. The authors have already shown that the mutants are deficient in antimicrobial activity, so to show the same thing in vivo is not very impactful. As another reviewer pointed out, the in vivo data provides no additional structural or mechanistic insight into LysIME-EF1, and I agree. I think the data should be removed. If the authors choose to keep the in vivo data, then many other issues will need to be addressed:

- In vivo, Issue 1. The authors did a pilot experiment to determine the minimum lethal dose (MLD) in lines 554-557. However, they never report what dose was experimentally determined to be the MLD. On line 353, it is stated that the mice were treated with 2 x 10^10 cfus, so one presumes this was the MLD, yet the figure legend (line 665) states the mice were treated with 4 x 10^9 cfus. Which is it? Also, the 4 x 10^9 dose was not one of the doses tested for MLD, so there is no rationale for how they derived this dose.

- In vivo, Issue 2. It is stated that mice were divided into 10 groups of 8 mice (lines 559-560). Figure 7 only shows 8 groups (WT, control, and 6 mutants). What is the purpose of the 2 remaining groups?

- In vivo, Issue 3. In both the original manuscript and the revised manuscript the authors consistently state that they used 8 mice for each group. However, in their rebuttal letter, they state they did the experiments in triplicate, indicating N=24 per group, not N=8 per group. This information is not stated anywhere in the manuscript. Were these new experiments since the original submission? If so, things need to be revised to state N=24 per group.

- In vivo, Issue 4. Statistics. Error bars are presented, which is odd in an in vivo experiment where normally the total N value is reported. As stated above, the authors never signify that experiments were done in triplicate, but this must be the case as they report 95.83% protection for WT (line 357), which cannot be derived from a group of 8 mice, but can be derived from a group of 24 mice (i.e. 23 out of 24 protected). They should report the data this way (i.e. 23/24). More importantly, the authors present no statistical data to compare results between groups, which is a significant flaw and must be addressed.

- In vivo, Issue 5. What is VREF (lines 549 and 665). This term is never defined.

- In vivo, Issue 6. The mice are monitored for 4 days according to the methods (lines 564-566). However, the authors state that the experiments were only 2 days in the results (line 356) and figure legend (line 667). Which is it?

- In vivo, Issue 7. Lines 354-356 are awkward. I think some sentences were combined or there is an extra period in line 355.

- In vivo, Issue 8. The results only provide life or death metrics. Was there any attempt to measure health status…i.e. even if a mouse is alive at day 2 or day 4 does not mean the mouse is healthy or otherwise will not die the next day, which is why these types of experiments should either last longer than a few days or should include some type of health monitoring matrix.

- In vivo, Issue 9. Graph. In the original manuscript, the authors presented the data in Kaplan Meier survival curves, which was appropriate. However, because several lines overlapped, they chose to present the data in a bar graph in the revised manuscript. This is not advisable because it loses all the temporal associated data displayed in a Kaplan Meier graph. I would go back to the KM graph even if multiple KM graphs are needed due to the overlap.

- Final conclusions. In lines 422-423 the authors state that residue R190 is essential for binding to the host cell wall and hence the biological activity of the endolysin. They then state in lines 424-26 that alteration of this residue could provide an opportunity to design endolysins with high specificity, efficacy, or broader spectrum. If R190 is essential for binding, it is not clear how alteration of this residue in particular can potentiate all of these actions. While I agree that the authors have described essential residues and mechanisms that increase our general understanding of endolysins and this general understanding may one day inform future engineering efforts, I think the comments specifically aimed at alteration of residue R190 without supporting data is a bit overstated.

- Turbidity reduction assays. In lines 521-528 the authors describe the general methods for the turbidity reduction assay. In the revised manuscript, all of the turbidity reduction assay experiments have been removed and replaced by the bacteriolytic assays. The one exception is the experiment with EDTA and metal ions presented in Figure 3D. Therefore, rather than general methods, these methods should specifically include experimental details for the EDTA/metal ion data. For example, specify how much EDTA was used, describe the dialysis steps, and describe the supplemental addition of metal ions.

- Entire manuscript. The entire manuscript should be better proofread by a native English speaker. This was a comment in the first review and the authors state in their cover letter that the language has been improved in the revised manuscript. However, this is not the case. Many language errors remain and some of the newly edited additions are even worse grammatically than the original version.

**Part III – Minor Issues: Editorial and Data Presentation Modifications**

Reviewer #1: The revised manuscript is improved with additional experiments and has addressed most of my previous concerns. However, the authors chose to keep the results on animal model, which is all right to show some additional data but there are some things to be clarified.

1. Bacterial dose. In the materials and methods section, the authors evaluated the virulence of E. faecalis 02 with different bacterial dosage, i.e., 2×107, 2×108, 2×109 and 2×1010 CFU/mouse, to seek the minimal lethal dose and the authors said the minimal lethal dose of bacteria was used in the animal infection experiments (page 25, line 449). While, in the results section, bacterial dosage is ≥2 × 1010 cfu/mouse (page 15, line 272). And in the corresponding Figure 7 legend, the injected bacterial dosage is 4×109 cfu/mouse (page 30, line 519). These data are not consistent. What is the minimal lethal dose of the strain tested?

2. In the materials and methods section, survival rates of the animals were recorded every 8 h for the first 48 h and then every 12 h for up to 4 days post-infection. The survival rates shown in Figure 7 are the survival rates at 2 days post-infection? Why did the authors choose 4 days of observation and present the survival rates on day 2?

3. In both methods section and figure legend, a number of n=8 animal/group was given. What do the error bars in Fig 7 stand for? Did the authors repeat the experiments (n=8 animal/group) for several times? If that is the case, the repeat times should be given accordingly. Also it looks odd for percentages of 96% mortality rate (Line 275), 95.83% (Line 276) and 16.66% (line 279) 54.17% (line 281), and 45.83% (line 282). How were these percentages calculated if the denominator is 8?

Minor points：

-Methods for Fig 1C and 1D, as well as Fig 5D etc were missing.

--Page 6, line 91, Fig S1 (p30, line 523), broader bactericidal spectrum. Please indicate which lysin LysIME-EF1 was compared to?

--page 15, lines 274-275, the sentence is confusing. Please change to “… variants (20 ug) 1 h post-infection. Mice that were not….”.

--page 20, site-directed mutagenesis. All primers used should be given in the supplementary.

--Line 416, turbidity reduction assay. It’s better to use g instead of rpm? Please check other places.

--page 24, line 424, OD600 should be shown as OD600.

--Line 428, PBS is not negative control, it is blank control.

--page 29, line 514 and line 516, delete the second LysIME-EF1.

--Page 30, line 519, what VREF stands for? A full name should be given when it was shown first time.

--Page 30, line 523, delete r between broader and bactericidal.

Reviewer #2: (No Response)

Reviewer #3: (No Response)

PLOS authors have the option to publish the peer review history of their article (what does this mean?). If published, this will include your full peer review and any attached files.

Reviewer #1: No

Reviewer #2: No

Reviewer #3: No
---

## [Editor Report · Decision Letter 2]

10 Feb 2020

Dear Prof. Ouyang,

We are pleased to inform you that your manuscript 'Structural and functional insights into a novel two-component endolysin encoded by a single gene in Enterococcus faecalis phage' has been provisionally accepted for publication in PLOS Pathogens.

Before your manuscript can be formally accepted you will need to complete some formatting changes, which you will receive in a follow up email. A member of our team will be in touch within two working days with a set of requests.

Best regards,

Gongyi Zhang

Associate Editor

PLOS Pathogens

Michael Otto

Section Editor

PLOS Pathogens

Kasturi Haldar

Editor-in-Chief

PLOS Pathogens

orcid.org/0000-0001-5065-158X

Michael Malim

Editor-in-Chief

PLOS Pathogens

orcid.org/0000-0002-7699-2064
---

## [Editor Report · Acceptance letter]

10 Mar 2020

Dear Prof. Ouyang,

We are delighted to inform you that your manuscript, "Structural and functional insights into a novel two-component endolysin encoded by a single gene in *Enterococcus faecalis* phage," has been formally accepted for publication in PLOS Pathogens.

Best regards,

Kasturi Haldar

Editor-in-Chief

PLOS Pathogens

orcid.org/0000-0001-5065-158X

Michael Malim

Editor-in-Chief

PLOS Pathogens

orcid.org/0000-0002-7699-2064